# Are people really less moral in their foreign language? Proficiency and comprehension matter for the moral foreign language effect in Russian speakers

Alena Kirova[1], Ying Tang[2], Paul Conway[3]*

1 Department of English and World Languages, Youngstown State University, Youngstown, Ohio, United States of America, 2 Department of Psychology, Youngstown State University, Youngstown, Ohio, United States of America, 3 School of Psychology, University of Southampton, Southampton, United Kingdom

* p.conway@soton.ac.uk

**Data Availability Statement:** As stated in the manuscript, Data and code for the study are available at https://osf.io/dvr5n/?view_only=

## Abstract

Previous work has demonstrated that people are more willing to sacrifice one person to save five in a foreign language (FL) than in their native tongue. This may be due to the FL either reducing concerns about sacrificial harm (deontological inclinations) or increasing concerns about overall outcomes (utilitarian inclinations). Moreover, proficiency in a foreign language (FL) may moderate results. To test these possibilities, we investigated the moral foreign language effect (MFLE) in a novel sample of Russian L1/English FL speakers. We employed process dissociation (PD)—a technique that independently assesses concerns about rejecting harm and maximizing outcomes in sacrificial dilemmas, and we assessed measures of objective and subjective foreign language proficiency and of dilemma comprehension. Results replicated the pattern of increased acceptance of sacrificial harm in FL demonstrated in earlier studies, but a PD analysis showed no evidence of increased concerns for utilitarian outcomes in a FL; instead, this pattern was driven by reduced concerns regarding sacrificial harm. However, people who reported better dilemma comprehension in the FL demonstrated both stronger deontological and utilitarian responding, and people with higher objective proficiency displayed stronger utilitarian responding in the FL than those with lower proficiency. These findings show that utilitarian inclinations are affected by reading dilemmas in a foreign language mainly in low-proficiency speakers, and that while emotional concerns for sacrifice are reduced in FL, better comprehension can increase such concerns as well as concern for outcomes.

## Introduction

Imagine you are a surgeon with five patients who will die without organ transplants. You have no spare organs, but you could terminate the life support of a long-time coma patient to save the other lives. Should you? Now, imagine you must resolve this scenario in a foreign language—does that affect your answer? Considerable research suggests that contemplating such

**Funding:** The author(s) received no specific funding for this work.

**Competing interests:** The authors have declared that no competing interests exist.

dilemmas in a foreign language increases acceptance of sacrificial harm, a phenomenon known as the *moral foreign language effect* (MFLE, e.g., [1]). This tendency has been explained in two ways: first, contemplating sacrificial harm in a foreign language results in blunted emotional reactions; second, the increased deliberative processing required when considering dilemmas in a foreign language promotes a focus on maximizing outcomes.

One way to distinguish between the mechanisms of blunted emotion and those of increased deliberation is to examine FL proficiency, given that the extent to which one's emotional reaction to harm is blunted and/or deliberation enhanced should be smaller among high-proficiency language learners than among those with lower proficiency. On the one hand, if the MFLE primarily reflects blunted harm rejection, high-proficiency learners should reject sacrificial harm more often than low-proficiency learners. On the other hand, if the MFLE primarily reflects increased deliberation, high-proficiency learners should maximize outcomes less often than low-proficiency learners do. We tested these possibilities by employing process dissociation (PD) to independently assess harm-rejecting and outcome-maximizing inclinations [2] in a sample of Russian native speakers who have different levels of English proficiency. To our knowledge, this is the first study that uses PD to directly investigate the role of language proficiency in order to test mechanisms underlying the MFLE.

## Sacrificial dilemmas

Sacrificial dilemmas entail a choice between harming a single target to maximize overall outcomes—e.g., killing the coma patient to save the other patients—or refusing such harm even if this means that a greater number of people will suffer. Such dilemmas originated in philosophy [3] but have become a common research tool for examining moral judgment. Rejecting sacrificial harm aligns with *deontological ethics* that judge the morality of an action by its nature (i.e., killing is wrong [4]), whereas accepting sacrificial harm (to maximize outcomes) aligns with *utilitarian ethics* that judge the morality of an action by its consequences (i.e., total lives saved [5]).

Although dilemma decisions may be *consistent with* philosophical positions, they are not *caused by* philosophical principles, but instead, reflect a complex array of psychological processes [6, 7]. According to the popular but controversial dual process model [8], harm-rejecting decisions reflect the dominance of emotional aversion to harm, whereas outcome-maximizing decisions reflect the dominance of cognitive deliberation regarding outcomes. Although the dual process model is often regarded as simplistic and has many critics (e.g., [9, 10]), a considerable body of evidence approximately aligns with dual process claims (e.g., [11–13]). Some work [14] suggests that the dual process model is not so much incorrect as incomplete, insofar as it describes only two of many processes relevant to sacrificial decision-making. It is therefore still useful to cautiously consider dual process model claims for interpreting the MFLE.

## Sacrificial dilemmas in a foreign language: MFLE research

The MFLE describes the tendency to accept sacrificial harm to maximize outcomes more often in a foreign language than in one's native tongue. The MFLE was originally discovered in [1] with studies of English/Spanish, Spanish/English, Korean/English, English/French, Spanish/Hebrew, and English/Hebrew speakers. Two potential explanations have been proposed, both drawing from the dual process model [15]. The dominant theoretical proposal is the *blunted deontology* account [1, 16], based on extensive research showing that speakers experience reduced emotionality in the FL compared to the first language (L1) (see [17, 18] for reviews of the research on emotionality as well as the research on embodied cognition in L1 vs. FL). This account posits that in-depth processing of evocative sacrificial scenarios in an L1 allows one to

powerfully visualize harm without necessarily shifting the degree of cognitive processing, resulting in decision-making that favors affective processes and in turn, an impulse to reject harm.

Conversely, confronting the same scenarios in one's FL is less evocative, as speakers cannot visualize the same harmful act without shifting their cognitive processing, resulting in reduced emotional reactions and thereby leading to a balance of mechanisms that favor cognitive processes and hence utilitarian responding. Such arguments align with research showing that people find profanity [19] and humor [20, 21] less evocative and lies (vs. truthful statements) less distressing [22] in FL than in L1.

A second account—the *heightened-utilitarianism* account [1, 16]—argues that the cognitive demands of processing information in a foreign language may force people to increase deliberation, which is associated with increased concerns for outcomes (e.g., [23, 24, cf. 25]). In this view, people tend to accept sacrificial harm in a foreign tongue more often than in their L1, not because they are less concerned about the prospect of causing harm in FL than in L1, but rather because more effortful cognitive processing in FL activates analytic processes and deliberative thinking, thereby increasing concerns about maximizing outcomes.

To date, neither hypothesis has been conclusively supported by evidence. They have mostly been tested using different versions of dilemmas—"personal" dilemmas where harm is physical and evocative, like pushing a person off a footbridge to save five, or "impersonal" dilemmas where harm is distant and mechanically mediated, like pressing a switch to kill one to save five [1, 26–29]. Studies generally show the MFLE in personal but not impersonal dilemmas, suggesting the MFLE reflects reduced emotional responses to sacrifice, supporting the blunted deontology account and challenging the heightened utilitarianism account. However, some studies suggest that ratings of emotionality of dilemmas do not mediate the MFLE on sacrificial responses [27, 30, 31], calling into question the blunted deontology account. At the same time, a recent study [32] found that the participants' emotions did mediate the MFLE in highly emotional dilemmas: their participants used high-arousal words to justify their decisions on the dilemmas considerably more frequently in their L1 than in their FL, and the L1 was associated with stronger deontological responding.

## The role of proficiency

Given the inconclusive findings from past work regarding the *blunted deontology* account and the *heightened utilitarianism* account, we examined the role of proficiency in the relevant foreign language to test the two hypotheses. Proficiency can be defined as an indicator of one's general ability in multiple components of language such as vocabulary, grammar, phonology, and pragmatics as well as sociolinguistic knowledge—the knowledge of rules for using language in society [33, 34]. Both the blunted deontology account and the heightened-utilitarianism account rely on people struggling to process either the emotional impact of actions or the general content of dilemmas, which should be primarily true of people with weak FL abilities.

Yet, although the MFLE has been tested in several studies, only a handful treat proficiency as a primary variable [27, 35–38], and findings from this research also remain inconclusive. For example, [37] found no proficiency effect in L1 Russian/FL English participants on the footbridge and crying baby dilemmas. By contrast, [27] found that lower proficiency was accompanied by higher acceptance of sacrificial harm on the footbridge dilemma and the lost wallet dilemma, but not on the trolley and crying baby dilemmas, in Chinese and German native speakers who learned English as their FL. Furthermore, one meta-analysis of the MFLE in decision-making found that proficiency did not affect the MFLE [35], while another [38] showed that higher self-reported reading proficiency was associated with an increased

likelihood of accepting sacrificial harm. Finally, a recent study [36], emphasizing that bilingualism is not a unitary but rather a multidimensional phenomenon, looked at the effect of bilingual language experience (age of acquisition, proficiency, and language dominance) on moral judgment. Among other things, the study found that utilitarian inclinations increased with proficiency in early bilinguals, but decreased in late bilinguals, while deontological inclinations decreased with proficiency in early bilinguals but increased with proficiency in late bilinguals.

## Challenges with interpreting conventional dilemma analyses

Taken as a whole, the corpus of previous studies—both on MFLE in general and on the role of proficiency in MFLE in particular—has generated inconclusive findings, a fact that may partially reflect the limitations of analyses that conflate the effects of blunted emotionality and enhanced reasoning. Most MFLE studies examine only sacrificial dilemmas where analyses contrast rejecting sacrificial harm (consistent with deontology) with accepting sacrificial harm for the greater good (consistent with utilitarianism), such as the crying baby dilemma where killing a baby saves a village. By treating these responses as inverse opposites, conventional analyses correlate stronger utilitarian responding with weaker deontological responding and vice versa, even though theory posits independence [8].

Therefore, the two explanations for the MFLE are conflated in conventional analyses: one cannot tell whether increased acceptance for sacrificial harm supports the blunted deontology or enhanced utilitarian models, as both patterns look the same in this analysis. Likewise, one cannot tell whether proficiency data support one or the other account because low-proficiency speakers should show stronger acceptance of harm than high-proficiency speakers do under both accounts. Moreover, treating harm-rejecting and outcome-maximizing inclinations as inverse opposites can lead to *suppression*—where results are "canceled out" when both tendencies point in the same direction. If, as some studies suggest [39], foreign language processing may reduce both deontological and utilitarian responding, these effects would negate each other in the conventional dilemma analysis. Suppression thus potentially explains a lack of significant MFLE in some studies (e.g., [30, 37]). To address these concerns, we employed PD to independently assess harm rejection and outcome-maximization response tendencies.

## Modeling approaches

To overcome the limitations of traditional analyses, researchers have employed modeling approaches such as PD [2]. Such modeling techniques are widely used in psychology (e.g., [40, 41]), and have been adapted to assess dilemma responses. Whereas traditional dilemmas involve only cases where causing harm saves lives—treating deontological and utilitarian responses as opposites—PD assesses both *incongruent dilemmas*, where causing harm maximizes outcomes, and *congruent dilemmas*, where causing harm arguably does not maximize outcomes. Thus, deontological and utilitarian responses are sometimes incongruent but other times congruent. For example, the incongruent surgeon dilemma involves deciding whether to terminate a coma patient's life support to save the lives of five people who require an emergency organ transplant—harm maximizes net lives. By contrast, the congruent surgeon dilemma involves deciding whether to sacrifice the same coma patient not to save lives but rather to simply reduce the effects of a long-term disease for five people—a harm which is arguably not "worth it."

By examining patterns of responses across cases, PD uses linear algebra to estimate the degree to which people a) reject sacrificial harm, regardless of outcomes (called the deontology parameter) and b) maximize outcomes, regardless of whether doing so requires rejecting harm

(called the utilitarian parameter)—note that these parameters reflect responses consistent with but not caused by their respective philosophical position. This requires measuring acceptance or rejection of sacrificial harm to congruent and incongruent dilemmas, and then computing the six equations described by Conway and Gawronski [2] to produce the deontological and utilitarian parameters (see S1 Appendix).

PD can help shed light on the theoretical underpinnings of the MFLE by clarifying whether making decisions in one's FL blunts deontological responding or increases utilitarian responding, or perhaps both or perhaps a pattern that is even more complex. PD can also reveal cases of suppression, where a given factor influences both parameters in the same direction.

## MFLE research using modeling approaches

Several studies have employed PD [16, 39] and related process models [42, 43] to investigate the possibility of suppression effects underlying the MFLE (see [25, 44]). Each of those studies has found evidence for weaker deontological responding in FL than in L1. In addition, all of them except [43, 44] found evidence for weaker utilitarian responding in FL than in L1. In most cases, the effect of foreign language processing on deontological and utilitarian responding was approximately equal in size, though in some cases, such as in [16], the decrease in deontological responding was more robust than the decrease in utilitarian responding.

The findings of reduced utilitarian responding in the foreign language condition observed in these studies effectively refutes the heightened utilitarianism account, which posits that processing in one's FL promotes deliberative thinking and increased utilitarian responding. In fact, to explain reduced utilitarian responding observed in their participants' FL, the authors in [16] hypothesized that FL overloads the mental capacities of a speaker and hence reduces deliberation, thereby affecting utilitarian responding. To test this hypothesis, they examined their participants' self-reports of FL proficiency and found tentative support for it in two of three of those experiments: The reduction in utilitarian responding was more prominent in participants with lower self-reported FL proficiency than in those with higher self-reported FL proficiency. Their study is the only one employing a modeling approach that has used proficiency to test a hypothesis concerning the MFLE.

Thus, in the previous research on the MFLE, blunted harm rejection associated with reduced emotionality in the FL as well as more effortful processing associated with processing dilemmas in one's FL have been proposed as two potentially overlapping explanations for the MFLE. If the MFLE is caused by a blunted emotional reaction to harm in the FL, it should depend on the amount of emotionality reduction in the FL, which should, in turn, depend on proficiency. Lower proficiency learners should have the highest amount of emotionality reduction since they are typically true *foreign* language learners who have had few opportunities to accumulate affective experiences in the FL. The term 'foreign language' (FL) refers to a language other than one's native tongue, learned in a formal classroom situation in a country *where the native language is dominant* [45]; for example, students learning Spanish in a university classroom in the USA are learning a foreign language.

Higher proficiency, however, is more likely achieved by complementing classroom instruction with interaction with native speakers and/or being exposed to authentic materials, leading to stronger emotional resonances in the FL [17, 18, 22, 46]. Higher proficiency language learners are, then, more likely to be *second* language learners, where the term 'second language' (L2) refers to any language learned after the first language in a formal classroom and/or in a naturalistic context (foreign language learners are thus a subset of second language learners); for example, Korean students learning English in the USA are second language learners. Therefore, as FL proficiency grows, so should emotional responses to harm, leading to generally

more affective responses to moral dilemmas, as manifested in higher deontological parameter scores.

Likewise, if the MFLE is a result of increased deliberation caused by cognitive processing in the FL, it should depend on the *extent* of the increase in deliberation. This increase may potentially affect the MFLE in two opposite ways. First, along the lines of the heightened utilitarianism account, greater processing demands in lower proficiency FL learners may promote more deliberative thinking and lead to higher utilitarian responding compared to those with higher proficiency. Alternatively, as [16] and [47] suggest, processing in an FL may act as a simultaneous processing burden, i.e., cognitive load, similar to remembering a phone number at the same time as reading a passage. According to this *cognitive overload* hypothesis, this additional cognitive load may overburden the mental capacities of a low-proficiency speaker and lead to weaker utilitarian responding in low-proficiency language learners compared to high-proficiency L2 speakers. This hypothesis is compatible with findings in Second Language Acquisition (SLA) research, which show that L2 processing in low-proficiency speakers is effortful and conscious, whereas it becomes more automatic for high-proficiency L2 speakers, for whom cognitive resources are spared for tasks other than processing the L2 (in this case, solving dilemmas) [48–50]. In summary, high proficiency should lead to increased deontological responding, consistent with the blunted deontology account, but it could lead to either reduced utilitarian responding (the cognitive overload hypothesis) or increased utilitarian responding (the heightened utilitarianism account). These are the possibilities we examined in our study.

## The current work

We employed PD to test the effect of proficiency on responses to moral dilemmas in a unique sample of Russian L1/English FL speakers. Participants completed the Conway and Gawronski moral dilemma battery [2] in either Russian or English. This procedure allowed us to calculate a) conventional sacrificial dilemma responses where causing harm maximizes outcomes (i.e., increased 'utilitarian versus deontological' responding), similar to most MFLE work (e.g., [1]), as well as b) PD parameters reflecting harm-rejection (deontological) and outcome-maximization (utilitarian) response tendencies across all dilemmas (e.g., [16]). We compared response patterns when people encountered dilemmas in their foreign versus native tongues, taking into account their FL proficiency.

We expected to replicate findings from the previous MFLE modeling studies, namely observing reduced deontological responding and potentially reduced utilitarian responding in L2 compared to L1 in this novel sample of Russian L1/English FL speakers. We anticipated a positive correlation between proficiency and both utilitarian and deontological responding, as more proficient FL speakers may be more sensitive to both harms and outcomes. We anticipated these findings would hold in regressions controlling for age, gender, and the other parameter (deontology or utilitarian parameter). Finally, we expected that language effects would be greatly reduced or even absent in the high-proficiency group, because high-proficiency should allow for automatic and emotional language processing in the FL and the L1.

Following the current trends in the SLA field [51], we used both subjective proficiency measures (self-ratings across the four language skills–reading, writing, listening, and speaking) and objective measures (a standardized language task–the Michigan Proficiency Test). However, recognizing the notoriously biased nature of self-assessment (e.g., [52, 53]), we expected the proficiency test to reveal a clearer effect of proficiency on MFLE than self-reports. At the same time, given that the language of dilemmas may differ substantially from the language used in general proficiency test questions, potentially undermining the utility of our proficiency test,

we also analyzed our participants' self-reported understanding of each dilemma. We followed APA ethical guidelines for the study. Data and code for the study are available at https://osf.io/dvr5n/?view_only=67e891c7ee66488e8d091aa78decd0c1.

## Method

### Participants

The study protocol was approved by the Youngstown State University Institutional Review Board. Electronic consent was obtained from all participants, who selected "I agree" or "I don't agree" after reading the consent form on the computer. Only participants who selected 'I agree' could proceed with the study. To preserve anonymity, their signatures were not collected. According to G-power, 96 participants would provide 95% power to detect a medium interaction term, $\eta_p^2 = .09$, in a $2 \times 2$ repeated measures ANOVA with language (Russian vs. English) as the between-subjects variable and parameter (utilitarian vs. deontological) as the within-subjects variable, assuming the parameters are correlated $r = .1$. We aimed to oversample, recruiting 353 L1 Russian/FL English students at two Russian higher institutions for partial course credit. We removed data from 116 participants for reasons including failure to complete the survey (100), reporting Vietnamese or Serbian as a first language (2), poor dilemma comprehension in English (reporting a score <5 on a 1–10 scale) in >70% of dilemmas (1), inability to calculate parameters due to a division by zero error (2), and low scores (0–20) on the objective proficiency measure (11). This left a final sample of 237 participants, 132 in the Russian ($M_{age} = 20.95$, $SD = 4.50$; Male = 21.2%, Female = 75.8%, unreported = 3%) and 105 in the English ($M_{age} = 21.24$, $SD = 5.35$; Male = 19.0%, Female = 81.0%) condition.

### Procedure

Participants were randomly assigned via Qualtrics to complete dilemmas in either Russian or English. Then they answered questions about their demographics and language learning history, rated their English skills, and took an English proficiency test.

### Materials

**Dilemma battery.** Participants completed a PD dilemma battery [2] in either Russian or English. This battery consists of 10 dilemmas, each with one incongruent and one congruent version, in a fixed random order. We conducted a conventional analysis of relative preference for utilitarian versus deontology responding, measuring the proportion of times people accepted (vs. rejected) sacrificial harm across the 10 incongruent dilemmas. We also computed PD parameters via the six equations described in [2], reflecting the tendency to reject sacrificial harm regardless of consequences (deontology parameter) and tendency to maximize outcomes regardless of causing harm (utilitarian parameter, see S1 Appendix).

The dilemmas were translated from English into Russian by a professional English/Russian translator, who is a native speaker of Russian. A native English-speaking professor of Russian back-translated them, after which both the original version and the back-translation were compared by an independent judge, a native speaker of English. Finally, a Russian-English bilingual reviewed the dilemmas in both languages to ensure that they matched conceptually.

**Demographics.** Participants reported age, gender, and language learning history: when they started learning English (age of acquisition), how long they had studied it (length of acquisition), whether they had lived in an English-speaking country, and whether they had relatives or close friends whose L1 was English (see S2 Appendix). On average, participants began learning English at age 7.27 ($SD = 2.76$) and studied it for 12.37 years ($SD = 4.22$).

**Dilemma comprehension.** Participants in the English condition rated the degree to which they comprehended each dilemma on a 10-point scale (1 = *I understood nothing at all*, 5 = *I understood the main idea*, 10 = *I understood everything*), Cronbach's α = .97. Mean reported comprehension was high, $M = 9.04$, $SD = 1.09$, range = 5.25–10.

**Self-rated English proficiency.** Participants reported English writing, reading, speaking, and listening proficiency on 5-point scales (1 = *beginner*, 5 = *advanced*), which were averaged into a self-reported proficiency score ($M = 4.02$, $SD = .61$, Cronbach's α = .86).

**Objective English proficiency.** Finally, participants completed 50 questions from the written portion of the Michigan Proficiency Test, a standardized proficiency test widely recognized as proof of English language ability ($M = 33.93$, $SD = 9.0$, on a scale of 0–50). This test consists of thirty multiple-choice questions that target grammar structures such as word order, tense, and prepositions, and twenty cloze questions, for which participants use context and vocabulary to identify the correct words that belong in the omitted texts of a passage (see S3 Appendix). Eleven participants scored low (0–20), 44 intermediate (21–34), and 61 advanced (35–50). As mentioned above, to ensure sufficiently meaningful comprehension of dilemmas, we removed data from the low-proficiency participants.

## Results

### Dilemma responding

We computed conventional and PD analyses.

**Conventional analysis.** We examined relative acceptance (vs. rejection) of sacrificial harm on incongruent dilemmas across the language condition. A univariate ANOVA controlling for age and gender showed that outcome-maximizing responses were significantly lower in participants' native Russian ($M = .42$, $SD = .26$) than FL English ($M = .50$, $SD = .26$), $F(1, 229) = 4.63$, $p = .033$. This pattern replicates most previous work the MFLE (e.g., [1]).

**Process dissociation analysis.** We computed an ANOVA on dilemma parameters with language (Russian vs. English) as a between-subjects variable and parameters (utilitarian vs. deontological) as a within-subjects variable, controlling for age and gender (see Fig 1). A trend toward overall higher scores in Russian ($M = .07$, $SD = .98$) than English ($M = -.09$, $SD = 1.02$) did not reach conventional statistical thresholds, $F(1, 229) = 3.07$, $p = .081$, $\eta_p^2 = .013$. There was also a theoretically uninteresting significant main effect of parameter, $F(1, 229) = 10.56$, $p = .001$, $\eta_p^2 = .044$, which emerged despite standardization due to the effect of covariates; this effect was eliminated in analyses without covariates. All other findings remained similar.

More importantly, the interaction was significant, $F(1, 229) = 4.38$, $p = .037$, $\eta_p^2 = .019$. Pairwise comparisons revealed significantly higher deontological responding in participants' native Russian ($M = .153$, $SD = .93$) than English ($M = -.193$, $SD = 1.05$), $M_{diff} = .346$, $SE = .129$, $p = .008$, Cohen's $d = .35$, whereas for utilitarian responding, there was no significant difference between Russian ($M = -.017$, $SD = 1.01$) and English ($M = .022$, $SD = .98$), $M_{diff} = -.039$, $SE = .13$, $p = .77$, Cohen's $d = .04$.

### Correlational analyses

Next, we examined correlations between all measures in the study (Table 1). Results revealed that objective and subjective proficiency correlated positively, $r = .44$, $p < .001$, meaning that those who perceived their English proficiency to be greater indeed fared better on the standardized English test. This correlation is not high, but it is similar to the coefficient ($r = .466$) reported in a meta-analysis on the relationship between self-reported language skills and externally measured abilities [54].

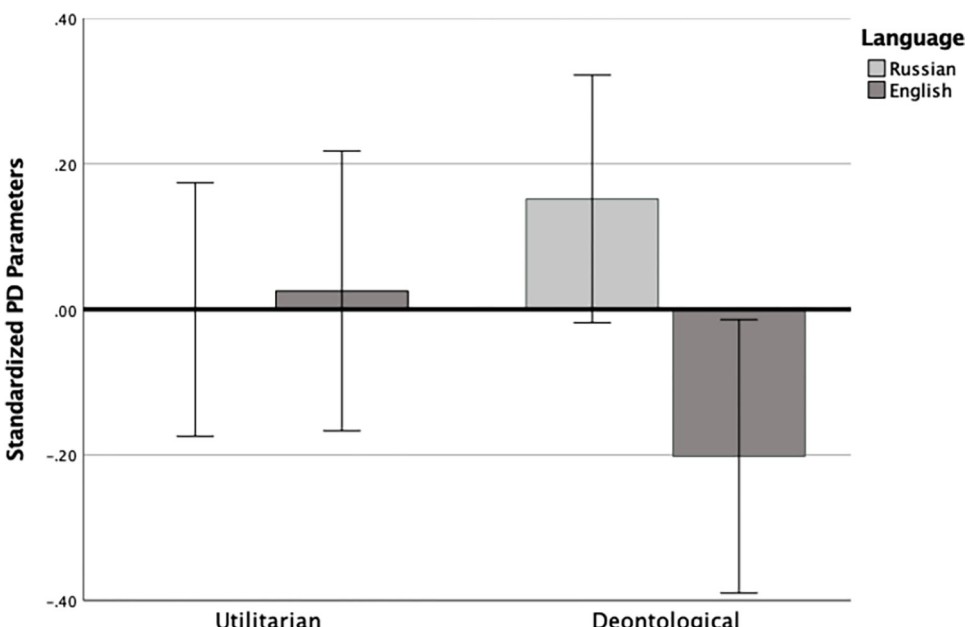

**Fig 1. Utilitarian and deontological process dissociation scores across native Russian (L1) and Foreign English Language (FL) conditions.** *Note.* Error bars reflect 95% CIs.

Surprisingly, objective English proficiency did not correlate significantly with conventional or PD dilemma scores, a result that is inconsistent with the proficiency hypothesis. Moreover, subjective proficiency correlated positively with utilitarian decisions in the conventional analysis, due to lower scores on the deontological PD parameter, but it did not correlate with the utilitarian parameter. However, these relationships did not hold in the subsequent regression analyses and may have been observed due to combining the two language conditions together in the correlation analysis.

## Regression analyses

**English proficiency and PD parameters.** Next, to test our hypothesis that higher proficiency in an FL/L2 increases deontological or utilitarian responding, we conducted a series of

**Table 1. Correlations between all variables in the study.**

| Variables | 1 | 2 | 3 | 4 | 5 | 6 |
|---|---|---|---|---|---|---|
| 1. Conventional Relative Utilitarian vs Deontological Judgments—i.e., Acceptance of Sacrificial Harm | - | | | | | |
| 2. Utilitarian PD Parameter | **.68***** | - | | | | |
| 3. Deontology PD Parameter | **-.67***** | .00 | - | | | |
| 4. Objective Proficiency | .08 | .13 | -.00 | - | | |
| 5. Subjective Proficiency | **.13*** | .06 | **-.14*** | **.44***** | - | |
| 6. Age | **-.24***** | **-.14*** | **.18**** | -.01 | -.06 | - |
| 7. Gender (1 = *m*, 2 = *f*) | -.07 | .01 | .10 | **.19**** | .07 | .04 |

Note.

* $p < .05$

** $p < .01$

*** $p < .001$

**Table 2. Regressing objective English proficiency and language condition on the utilitarian and deontological parameter scores.**

| Outcome Variable | Model | Predictor | B | SE | 95% CI | β | t | p |
|---|---|---|---|---|---|---|---|---|
| Utilitarian Parameter | Step 1[a] | Constant | **0.66** | **0.32** | **[0.02, 1.29]** | | **2.04** | **.043** |
| | | Language Condition[c] | -0.04 | 0.14 | [-0.33, 0.24] | -0.02 | -0.30 | .763 |
| | | Objective English Proficiency | 0.12 | 0.08 | [-0.03, 0.27] | 0.11 | 1.54 | .126 |
| | | Deontology Parameter Score | 0.03 | 0.07 | [-0.11, 0.16] | 0.03 | 0.38 | .706 |
| | | Age | **-0.03** | **0.01** | **[-0.06, -0.02]** | **-0.14** | **-2.11** | **.036** |
| | | Gender[d] | -0.03 | 0.17 | [-0.36, 0.30] | -0.01 | -0.18 | .860 |
| | Step 2[a] | Constant | 0.66 | 0.32 | [-0.13, 1.50] | | 1.67 | .097 |
| | | Objective English Proficiency × Language Condition | **0.40** | **0.18** | **[0.04, 0.75]** | **0.18** | **2.22** | **.028** |
| Deontology Parameter | Step 1[b] | Constant | **-0.80** | **0.31** | **[-1.42, -0.18]** | | **-2.55** | **.011** |
| | | Language Condition[c] | **-0.41** | **0.14** | **[-0.68, -0.14]** | **-0.20** | **-2.96** | **.003** |
| | | Objective English Proficiency | 0.08 | 0.08 | [-0.07, 0.23] | 0.08 | 1.08 | .282 |
| | | Utilitarian Parameter Score | 0.03 | 0.07 | [-0.10, 0.15] | 0.02 | 0.38 | .706 |
| | | Age | **0.04** | **0.01** | **[0.01, 0.07]** | **0.19** | **2.92** | **.004** |
| | | Gender | 0.20 | 0.16 | [-0.12, 0.52] | 0.08 | 1.23 | .221 |
| | Step 2[b] | Constant | **-0.79** | **0.31** | **[-1.41, -0.17]** | | **-2.52** | **.013** |
| | | Objective English Proficiency× Language Condition | 0.20 | 0.18 | [-0.15, 0.55] | 0.09 | 1.11 | .268 |

*Note.*

[a] $R^2_{Step1}$ = .031, $R^2_{Step2}$ = .051; $R^2_{Change}$ = .021, $p$ = .028.

[b] $R^2_{Step1}$ = .077, $R^2_{Step2}$ = .082; $R^2_{Change}$ = .005, $p$ = .268.

[c] Russian = 0, English = 1.

[d] Male = 0, female = 1. Bold indicates significance

regressions predicting each parameter from language condition and either objective or subjective proficiency, as well as the interaction between language and proficiency, controlling for age, gender, and the other parameter. We considered whether proficiency may have a curvilinear relationship with the parameters; however visual inspection of data suggested no such relationship; instead, linear regression was appropriate.

First, we predicted the utilitarian parameter from age, gender, deontology parameter, dummy coded language condition (0 = *Russian*, 1 = *English*), and standardized objective English proficiency scores at step 1, and the interaction between language and proficiency at step 2 (Table 2). The interaction between language condition and English proficiency scores was significant, showing that low-proficiency participants scored lower in utilitarian responding when reading dilemmas in foreign English than their native Russian; meanwhile, high-proficiency participants scored similarly in both languages.

We conducted a similar regression analysis on the deontology parameter controlling for the utilitarian parameter (Table 2). At step 1, language condition was significant, showing reduced deontological responding in FL. At step 2, the interaction between language condition and English proficiency scores was not significant: proficiency did not systematically increase participants' deontological inclinations in their foreign language.

We conducted similar regression analyses to test the relationship between subjective English proficiency, language condition, and deontological and utilitarian responses. None of the control variables or the covariates (age and gender) significantly predicted scores on the deontology and utilitarian parameters (Table 3).

**English language experience and PD parameters.** To test whether participants' age of acquisition (defined as the first exposure to the language) and length of learning English

**Table 3. Regressing subjective English proficiency and language condition on the utilitarian and deontological parameter scores.**

| Outcome Variable | Model | Predictor | B | SE | 95% CI | β | t | p |
|---|---|---|---|---|---|---|---|---|
| Utilitarian Parameter | Step 1[a] | Constant | 0.59 | 0.32 | [-0.05, 1.23] | | 1.83 | .069 |
| | | Language Condition[c] | 0.04 | 0.14 | [-0.23, 0.30] | 0.02 | 0.26 | .800 |
| | | Subjective English Proficiency | 0.05 | 0.07 | [-0.09, 0.18] | 0.05 | 0.69 | .494 |
| | | Deontology Parameter Score | 0.04 | 0.07 | [-0.10, 0.18] | 0.04 | 0.56 | .574 |
| | | Age | **-0.03** | **0.01** | **[-0.06, -0.01]** | **-0.14** | **-2.08** | **.039** |
| | | Gender[d] | 0.01 | 0.17 | [-0.32, 0.33] | 0.00 | 0.05 | .958 |
| | Step 2[a] | Constant | 0.57 | 0.32 | [-0.06, 1.21] | | 1.77 | .077 |
| | | Sub. English Proficiency × Language Condition | 0.22 | 0.15 | [-0.06, 0.51] | 0.12 | 1.54 | .126 |
| Deontology Parameter | Step 1[b] | Constant | **-0.83** | **0.31** | **[-1.45, -0.22]** | | **-2.68** | **.008** |
| | | Language Condition[c] | **-0.33** | **0.13** | **[-0.58, -0.08]** | **-0.16** | **-2.55** | ***.011*** |
| | | Subjective English Proficiency | -0.13 | 0.07 | [-0.26, 0.00] | -0.13 | -1.97 | .051 |
| | | Utilitarian Parameter Score | 0.04 | 0.06 | [-0.09, 0.16] | 0.04 | 0.56 | .574 |
| | | Age | **0.04** | **0.01** | **[0.01, 0.06]** | **0.18** | **2.76** | **.006** |
| | | Gender | 0.25 | 0.16 | [-0.07, 0.56] | 0.10 | 1.55 | .122 |
| | Step 2[b] | Constant | **-0.84** | **0.31** | **[-1.45, -0.23]** | | **-2.69** | **.008** |
| | | Sub. English Proficiency × Language Condition | 0.12 | 0.14 | [-0.16, 0.40] | 0.07 | 0.85 | .395 |

*Note.*
[a] $R^2_{Step1}$ = .022, $R^2_{Step2}$ = .032; $R^2_{Change}$ = .010, $p$ = .13.
[b] $R^2_{Step1}$ = .085, $R^2_{Step2}$ = .088; $R^2_{Change}$ = .003, $p$ = .395.
[c] Russian = 0, English = 1.
[d] Male = 0, female = 1. Bold indicates significance

moderated dilemma responding, we conducted regressions on each parameter parallel to those for proficiency. We found that neither age of acquisition, $b$ = .26, $p$ = .35, nor length of learning, $b$ = .18, $p$ = .34, predicted the utilitarian parameter. Similarly, neither age of acquisition, $b$ = -.29, $p$ = .31, nor length of learning, $b$ = -.15, $p$ = .41, predicted the deontology parameter.

**Dilemma comprehension and PD parameters.** Finally, we investigated whether people reporting better dilemma comprehension scored higher on the deontology and utilitarian parameters via regressions parallel to those for proficiency (though we could only examine people in the English condition for this analysis). We found that people reporting higher comprehension demonstrated both increased utilitarian responding, $b$ = .31, $p$ = .002, and increased deontological responding, $b$ = .27, $p$ = .011. Such findings suggest that the better people comprehend dilemmas, the more they tend both to reject sacrificial harm and maximize outcomes.

In sum, these results showed that self-assessed proficiency, age of acquisition, and length of learning did not interact with dilemma language to influence deontological or utilitarian responding. At the same time, externally measured (objective) proficiency interacted with dilemma language, with high-proficiency English learners showing stronger utilitarian responding. On the contrary, comprehension of dilemmas in the English condition predicted both deontological and utilitarian responding, with higher comprehension predicting stronger responding on both parameters (Table 4).

# Discussion

This study examined foreign language proficiency to clarify the process involved in the MFLE on sacrificial decision-making. The MFLE refers to the tendency of speakers to accept

**Table 4. Summary of regression results.**

| Predictor | Deontology Parameter | Utilitarian Parameter |
|---|---|---|
| **Age of Acquisition** (Interactions with Language condition) | No | No |
| **Length of Acquisition** (Interactions with Language condition) | No | No |
| **Subjective Proficiency** (Interactions with Language condition) | No | No |
| **Objective Proficiency** (Interactions with Language condition) | No | Yes |
| **Comprehension** (Main effects in the English Condition) | Yes | Yes |

sacrificial harm in order to maximize outcomes more often when they encounter dilemmas in their FL than in their native tongue. In other words, reading dilemmas in a foreign language seems to encourage responses that are consistent with utilitarian ethics but violate deontological ethics (e.g., [1]). Drawing on the dual process model of dilemma responding [15], theorists have explained this pattern as reflecting either blunted emotional reactions to sacrificial harm in a foreign language (the blunted deontology account) or increased deliberation caused by the extra cognitive demands required for reading dilemmas in a foreign language (the heightened utilitarianism account).

We employed FL proficiency in a sample of 237 Russian L1/English FL speakers with different levels of English proficiency to test the hypotheses: if the heightened utilitarianism account is correct, lower proficiency should be associated with increased utilitarian responding. If the blunted deontology account is correct, higher proficiency should be associated with increased deontological responding. Moreover, we used PD to independently assess the degree to which a foreign language impacts deontological and utilitarian responding.

Results partially confirmed the predictions. Consistent with previous studies on the MFLE, conventional analyses revealed that people provided more utilitarian and/or fewer deontological responses in a foreign language than in their native tongue (see Table 2). PD analyses clarified that this effect was driven by reduced deontological responding but not increased (or reduced) utilitarian responding in a foreign language—a pattern only partially consistent with other modeling work on the MFLE, which has shown lower deontological responding in a foreign language [43] while sometimes finding reduced utilitarian responding as well [16, 39, 42]. The current work also extends these findings to Russian speakers, a novel sample for this area.

### The role of proficiency in the MFLE

However, we also examined the role of foreign language proficiency, which revealed a somewhat different pattern. We found an interaction, such that participants low in objective language proficiency showed the opposite of the traditional MFLE: They scored lower in utilitarian responding in a foreign language than in their native tongue (see also Table 2). Conversely, a parallel regression for the deontology parameter replicated the main analysis, showing only a main effect of language such that people completing dilemmas in a foreign language scored lower in deontological responding than participants completing dilemmas in their native tongue. These results emerged while controlling for age, gender, and the other parameter. Hence, we found a pattern similar to some work [43] for high-proficiency participants (reduced deontological responding), whereas we found a pattern similar to other work [16, 39, 42] for low-proficiency participants (both reduced deontological and utilitarian responding). The current findings shed some possible light on why the patterns in previous studies are not

entirely consistent—it may be that some studies had higher proficiency foreign language speakers than others.

Together, these findings provide further support for the blunted emotionality account, suggesting that encountering dilemmas in a foreign language makes it particularly difficult to process elements that increase affective reactions to harm (e.g., [15, 55]). Similar to previous modeling research, these findings do not provide support for the heightened utilitarianism account, which argues that difficulty in processing FL increases cognitive processing in moral decision-making. Instead, they suggest that low-proficiency participants struggle to process the cognitive content of foreign language dilemmas, hence showing a deficit in utilitarian responding in addition to reduced deontological responding, supporting the cognitive overload hypothesis. It may be that reading dilemmas in a foreign language overburdens the processer and interferes with benefit-maximizing considerations in unskilled language learners, but not in skilled language learners, who process their FL more automatically (e.g., [55]). This finding is also consistent with research on the effect of proficiency on humor appreciation in FL vs. L1 (e.g., [21]).

Proficiency did not impact deontological responding, as both high and low-proficiency FL speakers demonstrated reduced deontological responding in their foreign language compared to their native tongue. This finding is consistent with the argument that deontological judgments partly reflect an affect-laden aversion to causing harm (e.g., [8, 15, 56]), and emotional processing is typically stronger in the native tongue than in the foreign language, due to the early age of acquisition of the L1 compared to the FL as well as the different ways the L1 and FL are learned. Despite a relatively early age of onset for English learning in our participants (around age 7), they likely did not learn their FL the way they learned their L1 (see the emotional context of learning hypothesis in [17]). Babies and very young children learn their native languages simultaneously with the development of emotional regulation systems [57]. They experience emotions as a reaction to specific physical or psychological sensations (sadness caused by pain, laughter caused by joy, etc.) at the same time as they are learning the words referring to these experiences. As a result, certain words become tightly interwoven with emotional experiences. Language learning in a foreign language classroom proceeds in a different manner and does not provide the learner with the full range of affective experiences; hence a language acquired in this way cannot evoke the same emotional reactions as the native tongue.

In addition, although our high-proficiency participants were skilled language speakers, they nonetheless were foreign language learners. They had lived in Russia their whole lives and thus naturally had had fewer and less rich affective experiences in English than in their L1, possibly leading them to exhibit weaker empathetic concerns about characters in dilemmas in the FL and eventually scoring lower on the deontology parameter. Thus, factors promoting *affective* experiences in the non-native language may be more relevant than proficiency for increasing one's deontological inclinations in that language. For example, immersion in the FL/L2 culture [29, 58, 59] or being an early bilingual [60–62] are shown to lead to an accumulation of affective experiences in both languages, decreasing the emotionality gap between L1 and L2.

Notably, these proficiency patterns emerged only for objective measures of proficiency rather than subjective measures, which did not significantly predict deontological or utilitarian responding (see Table 3). This may be due to the fact that subjective self-assessments are biased and fallible, especially for lower-skilled people who may fail to realize what they do not know —a pattern that has emerged in general meta-cognition (e.g., [52]) and second-language learning specifically [63, 64]. Alternatively, it may be that our subjective proficiency measure, which asked about reading, writing, listening, and speaking, did not capture performance on dilemmas as accurately as the objective measure, considering that dilemma performance involved only reading [51].

## The role of comprehension in the MFLE

Consistent with both the blunted affect and cognitive overload hypotheses, self-reported dilemma comprehension predicted increases in both the deontological and utilitarian parameters when participants encountered dilemmas in a foreign language. The better the participants understood the dilemmas in English, the stronger concerns they demonstrated both for rejecting sacrificial harm and maximizing overall benefits. This pattern suggests the MFLE may not be caused by the FL *per se*, but rather that reading in a FL leads to incomplete comprehension, which in turn leads to compromised moral responding. Given that L2 vocabulary knowledge is considerably restricted not just in breadth (how many L2 words one knows) but also in depth (how well one knows words) [65], this incomplete and superficial knowledge of semantic meanings may affect deep understanding of dilemmas and consequently affect people's moral decisions in their L2/FL. One can hardly experience strong emotional aversion to harm or a desire to maximize outcomes when one cannot fully understand the scale of harm or outcomes. These findings emphasize that in addition to taking into consideration bilingual language experience factors such as age of acquisition, proficiency, and language dominance [36], researchers also need to control for the extent to which specific experimental stimuli are understood by the FL readers.

## Implications for processing in foreign languages

Earlier research on the MFLE showed that people were more likely to endorse sacrificial harm in their FL than in their L1 and concluded that people were more "utilitarian" in their FL [1]. This was an inspiring finding for FL speakers and was quickly picked up by bilingual journalists working for popular news sources. For example, a Spanish native-speaking journalist writes in his L2 English for *BBC Worklife*, "Research has recently shown that people who can speak a foreign language are likely to be more analytical" and "if your work calls for a slow, rational and detached mind, using a foreign tongue gives you a small push" [66].

However, the current work parallels recent research on MFLE indicating that processing dilemmas in one's FL does not enhance utilitarian responding but instead can reduce deontological *and* sometimes also utilitarian responding. In addition, some studies have shown that processing in FL may be impaired in tasks that involve mental imagery [67], identifying logically invalid syllogisms [55] and 'fake news' [68], as well as judging hazards such as smoking and climate change [69].

These recent findings have also made their way into the popular press, with [70] publishing a piece with the provocative title "*Study*: *You are likely to make immoral decisions while speaking a second language.*" This is a problematic and potentially dangerous claim because it may bring harm to millions of already vulnerable immigrant populations who speak the country's majority language as their L2 (e.g., English in the U.S.). The issue of cognitive disadvantage in one's FL vs. L1 regarding decision-making is thus directly connected to the issues of bilingualism and immigration, making it a sensitive topic that demands due consideration.

While evidence for impaired decision-making in one's FL is growing, we emphasize that MFLE refers to a language effect in *FL* learners, who represent only a fraction of *bilingual/L2* speakers; whether this effect applies to other bilingual populations is an empirical question that future work should clarify. Crucially, our findings show that the roles of proficiency and comprehension should be emphasized in future research on L1 vs. FL decision-making. It is likely that the truth on the matter is nuanced and falls somewhere between "people who can speak a foreign language are likely to be more analytical" and "you are likely to make immoral decisions while speaking a second language."

## Limitations

As with all research, the current work has some limitations to consider. First, there is the issue of generalizability. Although we recruited people not often assessed in psychological work, we examined only two languages and two populations. Research has shown that cultural factors such as individualism, relational mobility (the extent to which a society allows its members to choose and/or dispose of interpersonal relationships), and religiosity have an effect on moral judgment [71], and Russian culture may differ considerably from both Western societies and Eastern societies such as South America or East Asia, where there may be different cultural expectations regarding dilemma responding or different patterns of language use. In fact, [72] showed that Russian participants differ systematically from participants from Western societies such as the US, Canada, and Britain, in that they tend to accept sacrificial harm less often. Thus, it remains unclear how well the current findings would be replicated in other cultures both in the East and the West, and more cross-cultural research is needed to address the issue of generalizability. That said, [16] found a somewhat similar pattern across six languages, suggesting results may not be limited only to the current sample.

Second, dilemma research has been criticized for employing hypothetical, artificial, and sometimes implausible scenarios (e.g., [73–75]). At the same time, the goal of research on moral dilemmas is not to document what people would do in real life, but rather to provide tight experimental control over the examination of situations otherwise impossible to measure ethically. Indeed, this body of work has illuminated systematic patterns of moral decision-making that cohere in a nomological network with a wide range of theoretical variables [76]. The current dilemmas, while more plausible than other more extreme examples (e.g., that a human body would stop a moving train), nonetheless do not necessarily portray entirely realistic situations. However, the PD parameters uniquely align with conceptually relevant variables such as empathic concern, cognitive reflection, moral identity, and psychopathy in theoretically expected ways, allowing for interpretations of the psychological processes involved in dilemma responses (e.g., [7]). Therefore, implausibility cannot explain differences across language condition, insofar as all participants encounter the exact same set of dilemmas, and hence any degree of implausibility should affect all participants similarly. As such, though the results of the current study rely on assumptions similar to those of most dilemma research, they nonetheless remain informative regarding the psychological processes involved in L2 language processing.

Third, PD is the simplest model in a family of models that parameterize dilemma responding. Recently, researchers developed a CNI model similar to the PD model employed here, except that it examines scenarios where the well-being of an individual is at odds with a group, and prioritizing the individual over the group requires either action or inaction [25]. The CNI model uses multinomial modeling to estimate parameter and model fit and allows for estimating sensitivity to *consequences* (comparable to the utilitarian parameter in PD), sensitivity to *norms* (somewhat comparable to the deontology parameter in PD), and general preferences for *inaction* (refusal to act regardless of outcomes or whether action will save lives). Inaction is confounded with the PD parameters because utilitarian responding sometimes requires action whereas deontological responding always requires inaction. Likewise, [44] developed a variant called the proCNI model similar to CNI, except that it focuses only on proscriptive scenarios, instead of varying whether the actor will continue or abort a course of action that will cause sacrificial harm.

Future work may profit from employing the CNI or proCNI models to determine if the pattern of results demonstrated here loads uniquely on concerns for the individual and group, or whether some of the variance may be due to systematic differences in inaction or inertia

tendencies in L2. That said, [42] discovered no effect of language on general action tendencies using the CNI model, instead finding that L2 reduced the norms and consequences parameters only. Conversely, using the proCNI model, [43] found evidence of reduced inertia—that is, a greater willingness to go along with the behavioral default—in a FL, but only in high-involvement scenarios (e.g., using a pillow to strangle someone rather than ordering a nurse to give them sleeping pills). In addition, in line with the current findings, they found that L2 reduced concerns consistent with deontological norms, though they uncovered no evidence of reduced concerns for utilitarian consequences in L2. Either way, these results suggest it is unlikely that the effect of L2 on reduced deontological inclinations is accounted for entirely by inaction or inertia.

Fourth, although we assumed reduced emotionality in FL compared to L1, we did not directly measure emotionality induced by reading dilemmas in FL vs. L1, relying instead on research in the field of language emotionality. Moreover, although we use the term "emotionality," we recognize that the definition of emotion is complex and there is no field-wide consensus regarding its usage at this point [77, 78]. No study on the MFLE has measured emotionality using objective measures so far, and our usage of the term "emotionality" reflects conventional use of it in the MFLE field and research investigating emotionality in language (for a review, see [20]). We recognize, however, that a lack of direct measures of emotionality in our study means that our conclusions regarding the role of emotionality in blunting harm-rejecting inclinations should be interpreted with caution. Future research should incorporate such measures to examine this possibility directly.

## Future directions

Some extant research on the effect of language on moral judgment in traditional high-conflict dilemmas has explored bilingual populations other than FL learners, such as early bilinguals [31] and proficient, highly acculturated L2 learners [29, 79, 80]. Future dilemma research would benefit from using modeling approaches such as PD, the CNI model, or the proCNI model to study such populations. Moreover, strengthening such research with objective measures of emotionality commonly used in research on language emotionality—such as skin conductance response and eye-tracking [60, 81]—may indeed reveal that deontological responding is equally strong in languages with similar emotional resonances to the dilemmas.

The weaker utilitarian responding in low-proficiency but not in high-proficiency learners found in this study is consistent with the cognitive overload hypothesis–the idea that processing dilemmas in FL by unskilled language speakers, whose language processing is effortful and lacks automaticity, leaves insufficient cognitive resources for dilemma processing, particularly with respect to benefit-maximizing inclinations. Future research would benefit from measuring the amount of cognitive load: if cognitive load is reliably higher in a low-proficiency than in a high-proficiency group during dilemma administration, and utilitarian responding is weaker in the low-proficiency group, one can conclude that lower proficiency is in fact related to the increase in cognitive load, affecting moral judgment. Some objective measures of cognitive load include pupillometry (eye-tracking), brain activity measures such as MRI and fNIRS, EEG or cardiovascular metrics, while subjective measures include self-reports of stress or mental effort [82].

One of the contributions this study makes to the MFLE field is that it potentially brings together and reconciles the seemingly contradictory findings from the other modeling studies–those in [43], which found weaker deontological but not utilitarian responding in the FL, vs. those in [42], which found that FL compromised both deontological and utilitarian responding. This discrepancy might have been caused by different proficiency levels in the FL

participants in those studies, and it is therefore appropriate to conduct modeling studies that will include proficiency as one of the independent variables, ideally using different language groups. That said, the different findings in those studies might also be related to the differences between dilemmas used in the PD/CNI vs. proCNI models.

Most importantly, the emphasis in research on the MFLE so far has been on identifying the psychological underpinnings of the effect. We suggest that linguistic variables, both in terms of the linguistic structures used in the dilemmas *per se* and considerations relevant for the capability of second/foreign language learners to comprehend dilemmas, need to be highlighted and controlled for as well in future research. For example, given the research on the L2 acquisition of semantic meanings mentioned above [65], controlling for the depth of semantic knowledge of the of the words used in moral dilemmas could be elucidating, especially considering that this study reveals that dilemma comprehension plays an important role in dilemma responding. Moreover, it would be informative to investigate the relationship between working memory capacity and the MFLE, given that Second Language Acquisition research has shown that many differences between L1 and L2 processing are caused by a lower working memory capacity in the L2 [83]. It is possible that the MFLE is also (at least partially) caused by it: while they are reading dilemmas in their FL, people may be unable to keep all the details of the dilemma in their working memory, potentially forgetting some important details by the time they make a decision. Finally, it may be that the learning conditions under which people encounter their FL influence responses, such as humanistic learning environments [84].

## Conclusion

In conclusion, we replicated and clarified previous work on the MFLE in a unique sample of Russian L1/English FL speakers. Replicating past work, we found that participants randomly assigned to complete dilemmas in their FL instead of L1 made more utilitarian judgments, but we did not find evidence for increased utilitarian concerns in FL—instead, a PD analysis revealed clear evidence for the blunted deontology account. This finding suggests that completing dilemmas in a foreign language makes it harder to process the emotional content involved in sacrificial harm, making sacrifice less aversive.

At the same time, we found evidence that FL proficiency mattered: people who scored higher on our standardized proficiency test in FL also scored higher in utilitarian concerns in their foreign language, suggesting that ease of processing due to high-proficiency enabled participants to fully appreciate the utility of sacrificial action and select answers that maximize outcomes. Furthermore, people who reported better dilemma comprehension in FL demonstrated increased deontological and utilitarian responding: These findings suggest that comprehending dilemmas in a foreign language increases both emotional concern for sacrifices and logical processing of the outcomes.

## Supporting information

**S1 Appendix. Process dissociation calculations.**
(DOCX)

**S2 Appendix. Linguistic background questionnaire.**
(DOCX)

**S3 Appendix. MELICET test adapted.**
(DOCX)

## Acknowledgments

The authors thank Mark Schaller for authoring JavaScript code used in the collection of data on Qualtrics, and Steven Reale and John Sarkissian for helping with proofreading and editing. The authors also thank Milana Khachaturova, Sergey Lee, and Alina Nakhodkina for helping with data collection, as well as Melissa Smith, John Sarkissian, Galina Machaeva, and Elena Brandt for help with translation and preparation of study materials.

## Author Contributions

**Conceptualization:** Alena Kirova.

**Formal analysis:** Ying Tang.

**Investigation:** Alena Kirova.

**Methodology:** Alena Kirova, Paul Conway.

**Project administration:** Alena Kirova.

**Supervision:** Paul Conway.

**Writing – original draft:** Alena Kirova.

**Writing – review & editing:** Alena Kirova, Ying Tang, Paul Conway.

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
