## [Decision Letter · Decision Letter 0]

27 Apr 2023

PONE-D-23-05151Are People Really Less Moral in Their Foreign Language? Proficiency and Comprehension Matter for the Moral Foreign Language Effect in Russian SpeakersPLOS ONE

Dear Dr. Conway,

Thank you for submitting your manuscript to PLOS ONE. After careful consideration, we feel that it has merit but does not fully meet PLOS ONE’s publication criteria as it currently stands. Therefore, we invite you to submit a revised version of the manuscript that addresses the points raised during the review process.

Dear authors, Thank you for your submission to PLOS ONE. All in all, you did a good job; however,  there are minor points,  as raised by reviewer 2, which should be addressed. Good luck. ==============================

We look forward to receiving your revised manuscript.

Kind regards,

Ehsan Namaziandost

Academic Editor

PLOS ONE

2. Please provide additional details regarding ethical approval in the body of your manuscript. In the Methods section, please ensure that you have specified the name of the IRB/ethics committee that approved your study.

Reviewers' comments:

Reviewer's Responses to Questions

**Comments to the Author**

1. Is the manuscript technically sound, and do the data support the conclusions?

Reviewer #1: Yes

Reviewer #2: Yes

2. Has the statistical analysis been performed appropriately and rigorously? 

Reviewer #1: Yes

Reviewer #2: Yes

3. Have the authors made all data underlying the findings in their manuscript fully available?

Reviewer #1: Yes

Reviewer #2: Yes

4. Is the manuscript presented in an intelligible fashion and written in standard English?

Reviewer #1: Yes

Reviewer #2: Yes

5. Review Comments to the Author

Reviewer #1: The current paper investigates the forein language effect on moral judgment. Prior research has uncovered that people are more likely to condone sacrificial choices on sacrificial harm dilemmas when they are asked to read those dilemmas in a foreign language vs when they are asked to read those dilemmas in their native language. The authors use process dissociation, a method to independently measure people's inclination towards both types of responses in these dilemmas so that they can investigate which of the two underlying inclinations drives the effect, and test whether foreign language proficiency moderates the effect.

The manuscript is very well written. It is clear, it discusses the relevant literature, demonstrates how the current work fits in with prior work and explains why the research question is relevant. The methods are appropriate and I appreciate that the authors included both a subjective and an objective measure of language proficiency. They gathered a sufficiently large sample, used an open science approach and their analyses are conducted the way they should be conducted. Their conclusions are supported by the data, and the authors take enough time to discuss the nuances in their data. Their discussion of the limitations of their data seems good to me as well.

Honestly, I don't have much to add to this paper as a reviewer. This is a straightforwardly well conducted study that has been written up appropriately. I think the authors did a great job and this manuscripts fits well within the goals and aims of Plos ONE. I truly don't do this often but I would be happy recommending it for publication as is.

Reviewer #2: This paper shows a valuable contribution to the existing literature and knowledge in terms of introduction, style of writing, and methodology. In addition, the manuscript shows very good academic language and interesting cohesion of ideas in the introduction analysis of results, and discussion. The title is very good and has a degree of innovation as not much research dealt with such subject. The analysis of the results is also very good and has been written in a good way.

Yet, there are minor typos that should be corrected.

1. The researchers need to add some more recommendations for further utilization of these findings and future lines of research could be also helpful as the research was conducted in a specific region and it is important to analyse if the situation will be different in other countries or the same.

2. Are the figures adapted or constructed? Clarify please.

3.Provide some more ideas on how the research could be extended and followed up, i.e., what are the further steps that need to be followed.

4.It still needs thorough proofreading and editing to polish its content, language, and a referencing style.

5. The significance of the study needs to be highlighted more.

6. Add more details on the application of the ethical consideration.

7. Are the figures adapted or constructed? Clarify please.

6. PLOS authors have the option to publish the peer review history of their article (what does this mean?). If published, this will include your full peer review and any attached files.

Reviewer #1: No

Reviewer #2: No

---

## [Author Response · Author response to Decision Letter 0]

16 May 2023

Note: This information duplicates the response to the editor and reviewers in the cover letter of this resubmission. 

Dear Dr Namaziandost, 

Thank you and the reviewers for this helpful and constructive feedback on PONE-D-23-05151, Are People Really Less Moral in Their Foreign Language? Proficiency and Comprehension Matter for the Moral Foreign Language Effect in Russian Speakers. 

We are pleased to hear that the reviewers saw merit in the manuscript, and we have made revisions in accordance with instructions to bring the manuscript into alignment with PLOS One guidelines and standards. Accordingly, we have uploaded a Revised Manuscript with Track Changes, as well as an unmarked version of the original manuscript. We also respond to each point below. 

COMMENT 

RESPONSE 

We have edited the paper to align with PLOS ONE style requirements, including naming our files accordingly, and have formatted author information accordingly. 

COMMENT 

2. Please provide additional details regarding ethical approval in the body of your manuscript. In the Methods section, please ensure that you have specified the name of the IRB/ethics committee that approved your study.

RESPONSE 

We have enclosed a copy of ethics approval with the study documents. 

We now specify in the introduction that “We followed APA ethical guidelines for the study.” We also clarified the full name of the IRB board in the methods section as follows: “The study protocol was approved by the Youngstown State University Institutional Review Board."

COMMENT 

RESPONSE 

We now clarify in the Method section the full name of the IRB board in the methods section as follows: “The study protocol was approved by the Youngstown State University Institutional Review Board. Electronic consent was obtained from all participants, who selected "I agree" or "I don't agree" after reading the consent form on the computer. Only participants who selected ‘I agree’ could proceed with the study. To preserve anonymity, their signatures were not collected.” 

COMMENT 

RESPONSE 

We have reviewed the reference list and checked it is correct and complete. Due to adding new paragraphs and sections in accordance with the reviewers’ comments, we added more citations, thereby extending our reference list to include the following eight additional references: 

1. Peñarredonda, J. The Huge Benefits of Working in Your Second Language. London: BBC Worklife. 2018 [cited 2023 May 5]. Available from: https://www.bbc.com/worklife/article/20180525-why-using-a-foreign-language-could-make-you-better-at-work

2. Hayakawa S, Keysar B. Using a foreign language reduces mental imagery. Cognition. 2018 Apr 1;173:8-15. http://dx.doi.org/10.1016/j.cognition.2017.12.010

3. Muda R, Pennycook G, Hamerski D, Białek M. People are worse at detecting fake news in their foreign language. Journal of Experimental Psychology: Applied. 2023 May 8. http://dx.doi.org/10.1037/xap0000475

4. Hadjichristidis C, Geipel J, Savadori L. The effect of foreign language in judgments of risk and benefit: The role of affect. Journal of Experimental Psychology: Applied. 2015 Jun;21(2):117.

5. Study: You are likely to make immoral decisions while speaking a second language, study finds. The Language Nerds. 2020 [cited 2023 May 5]. Available from: https://thelanguagenerds.com/how-morality-changes-in-a-foreign-language/

6. Awad E, Dsouza S, Shariff A, Rahwan I, Bonnefon JF. Universals and variations in moral decisions made in 42 countries by 70,000 participants. Proceedings of the National Academy of Sciences. 2020 Feb 4;117(5):2332-7.

7. Arutyunova KR, Alexandrov YI, Hauser MD. Sociocultural influences on moral judgments: East–west, male–female, and young–old. Frontiers in psychology. 2016 Sep 5;7:1334.

8. Martin S. Measuring cognitive load and cognition: metrics for technology-enhanced learning. Educational Research and Evaluation. 2014 Nov 17;20(7-8):592-621.

COMMENT 

Reviewer #1: The current paper investigates the forein language effect on moral judgment. Prior research has uncovered that people are more likely to condone sacrificial choices on sacrificial harm dilemmas when they are asked to read those dilemmas in a foreign language vs when they are asked to read those dilemmas in their native language. The authors use process dissociation, a method to independently measure people's inclination towards both types of responses in these dilemmas so that they can investigate which of the two underlying inclinations drives the effect, and test whether foreign language proficiency moderates the effect.

The manuscript is very well written. It is clear, it discusses the relevant literature, demonstrates how the current work fits in with prior work and explains why the research question is relevant. The methods are appropriate and I appreciate that the authors included both a subjective and an objective measure of language proficiency. They gathered a sufficiently large sample, used an open science approach and their analyses are conducted the way they should be conducted. Their conclusions are supported by the data, and the authors take enough time to discuss the nuances in their data. Their discussion of the limitations of their data seems good to me as well.

Honestly, I don't have much to add to this paper as a reviewer. This is a straightforwardly well conducted study that has been written up appropriately. I think the authors did a great job and this manuscripts fits well within the goals and aims of Plos ONE. I truly don't do this often but I would be happy recommending it for publication as is.

RESPONSE 

We thank the reviewer for these kind words and very much appreciate these thoughts. 

COMMENT 

Reviewer #2: This paper shows a valuable contribution to the existing literature and knowledge in terms of introduction, style of writing, and methodology. In addition, the manuscript shows very good academic language and interesting cohesion of ideas in the introduction analysis of results, and discussion. The title is very good and has a degree of innovation as not much research dealt with such subject. The analysis of the results is also very good and has been written in a good way.

Yet, there are minor typos that should be corrected.

RESPONSE 

We thank the reviewer for these encouraging comments. We have gone through the paper to correct typos. 

COMMENT 

1. The researchers need to add some more recommendations for further utilization of these findings and future lines of research could be also helpful as the research was conducted in a specific region and it is important to analyse if the situation will be different in other countries or the same.

RESPONSE 

The paper states, “Although we recruited people not often assessed in psychological work, we examined only two languages and two populations.” We expanded this statement by adding, “Research has shown that cultural factors such as individualism, relational mobility (the extent to which a society allows its members to choose and/or dispose of interpersonal relationships), and religiosity have an effect on moral judgment [71], and Russian culture may differ considerably from both Western societies and Eastern societies such as South America or East Asia, where there may be different cultural expectations regarding dilemma responding or different patterns of language use. In fact, [72] showed that Russian participants differ systematically from participants from Western societies such as the US, Canada, and Britain, in that they tend to accept sacrificial harm less often. Thus, it remains unclear how well the current findings would be replicated in other cultures both in the East and the West, and more cross-cultural research is needed to address the issue of generalizability.”

COMMENT 

2. Are the figures adapted or constructed? Clarify please.

RESPONSE 

The figure was constructed for this paper on the basis of the data in the study. 

COMMENT 

3. Provide some more ideas on how the research could be extended and followed up, i.e., what are the further steps that need to be followed.

RESPONSE 

Thank you for this suggestion. The paper states: 

1. “While evidence for impaired decision-making in one’s FL is growing, we emphasize that MFLE refers to a language effect in FL learners, who represent only a fraction of bilingual/L2 speakers; whether this effect applies to other bilingual populations is an empirical question that future work should clarify.” 

2. “Thus, it remains unclear how well the current findings would be replicated in other cultures both in the East and the West, and more cross-cultural research is needed to address the issue of generalizability.” 

3. “Future work may profit from employing the CNI or proCNI models to determine if the pattern of results demonstrated here load uniquely on concerns for the individual and group, or whether some of the variance may be due to systematic differences in inaction or inertia tendencies in L2.” 

4. “We recognize, however, that a lack of direct measures of emotionality in our study means that our conclusions regarding the role of emotionality in blunting harm-rejecting inclinations should be interpreted with caution. Future research should incorporate such measures to examine this possibility directly.” 

5. “Future dilemma research would benefit from using modeling approaches such as PD, the CNI model, or the proCNI model to study such populations. Moreover, strengthening such research with objective measures of emotionality commonly used in research on language emotionality—such as skin conductance response and eye-tracking [60, 75]—may indeed reveal that deontological responding is equally strong in languages with similar emotional resonances to the dilemmas.” 

We also added, “Future research would benefit from measuring the amount of cognitive load: if cognitive load is reliably higher in a low-proficiency than in a high-proficiency group during dilemma administration, and utilitarian responding is weaker in the low-proficiency group, one can conclude that lower proficiency is in fact related to the increase in cognitive load, affecting moral judgment. Some objective measures of cognitive load include pupillometry (eye-tracking), brain activity measures such as MRI and fNIRS, EEG or cardiovascular metrics, while subjective measures include self-reports of stress or mental effort [82].”

COMMENT 

4. It still needs thorough proofreading and editing to polish its content, language, and a referencing style.

RESPONSE 

We have edited the paper to fix errors and adapted it to PLOS ONE referencing and style guidance. 

COMMENT 

5. The significance of the study needs to be highlighted more.

RESPONSE 

The paper states in the discussion, “The current work also extends these findings to Russian speakers, a novel sample for this area” and “the current findings shed some possible light on why the patterns in previous studies are not entirely consistent—it may be that some studies had higher proficiency foreign language speakers than others.” These are two novel aspects of the current work. 

We have emphasized the latter novel aspect in the Future Directions section by saying, “One of the contributions this study makes to the MFLE field is that it potentially brings together and reconciles the seemingly contradictory findings from the other modeling studies – those in [43], which found weaker deontological but not utilitarian responding in the FL, vs. those in [42], which found that FL compromised both deontological and utilitarian responding. This discrepancy might have been caused by different proficiency levels in the FL participants in those studies, and it is therefore appropriate to conduct modeling studies that will include proficiency as one of the independent variables, ideally using different language groups.” 

Furthermore, the paper states, “This pattern suggests the MFLE may not be caused by the FL per se, but rather that reading in a FL leads to incomplete comprehension, which in turn leads to compromised moral responding” and “These findings emphasize that in addition to taking into consideration bilingual language experience factors such as age of acquisition, proficiency, and language dominance [36], the extent to which specific experimental stimuli are understood by the FL readers also needs to be controlled for.” 

Finally, we have added a section in the discussion on Implications for processing in foreign languages that connects the current findings to a broader discussion in the scientific and popular literature about the role of FL processing. 

COMMENT 

6. Add more details on the application of the ethical consideration.

RESPONSE 

We now clarify in the Method section the full name of the IRB board in the methods section as follows: “The study protocol was approved by the Youngstown State University Institutional Review Board. Electronic consent was obtained from all participants, who selected "I agree" or "I don't agree" after reading the consent form on the computer. Only participants who selected ‘I agree’ could proceed with the study. To preserve anonymity, their signatures were not collected.” 

COMMENT 

7. Are the figures adapted or constructed? Clarify please.

RESPONSE 

The figure was constructed for this paper on the basis of the data in the study.

---

## [Decision Letter · Decision Letter 1]

6 Jun 2023

PONE-D-23-05151R1Are People Really Less Moral in Their Foreign Language? Proficiency and Comprehension Matter for the Moral Foreign Language Effect in Russian SpeakersPLOS ONE

Dear Dr. Conway,

Thank you for submitting your manuscript to PLOS ONE. After careful consideration, we feel that it has merit but does not fully meet PLOS ONE’s publication criteria as it currently stands. Therefore, we invite you to submit a revised version of the manuscript that addresses the points raised during the review process.

We look forward to receiving your revised manuscript.

Kind regards,

Ehsan Namaziandost

Academic Editor

PLOS ONE

Journal Requirements:

Reviewers' comments:

Reviewer's Responses to Questions

**Comments to the Author**

1. If the authors have adequately addressed your comments raised in a previous round of review and you feel that this manuscript is now acceptable for publication, you may indicate that here to bypass the “Comments to the Author” section, enter your conflict of interest statement in the “Confidential to Editor” section, and submit your "Accept" recommendation.

Reviewer #2: All comments have been addressed

2. Is the manuscript technically sound, and do the data support the conclusions?

Reviewer #2: Yes

3. Has the statistical analysis been performed appropriately and rigorously? 

Reviewer #2: Yes

4. Have the authors made all data underlying the findings in their manuscript fully available?

Reviewer #2: (No Response)

5. Is the manuscript presented in an intelligible fashion and written in standard English?

Reviewer #2: Yes

6. Review Comments to the Author

Reviewer #2: This paper shows a valuable contribution to the existing literature and knowledge in terms of introduction, style of writing, and methodology. Yet, there are minor typos that should be corrected.

The manuscript should be checked for minor language issues and the researchers need to mention to some tables in the explanations so that they are clearer and easier to be followed. Furthermore, The discussion of the study needs to be compared with more relevant new studies in relation to that period such as:

Liqaa Habeb Al-Obaydi (2021): Humanistic learning elements in a blended

learning environment: a study in an EFL teaching context, Interactive Learning Environments, DOI:

10.1080/10494820.2021.1919717

Finally,it still needs thorough proofreading and editing to polish its content, language, and APA style.

The manuscript looks much better. Thank you.

7. PLOS authors have the option to publish the peer review history of their article (what does this mean?). If published, this will include your full peer review and any attached files.

Reviewer #2: No

---

## [Author Response · Author response to Decision Letter 1]

6 Jun 2023

Dear Dr Namaziandost, 

Thank you and the reviewers for this helpful and constructive feedback on this revision, PONE-D-23-05151R1, Are People Really Less Moral in Their Foreign Language? Proficiency and Comprehension Matter for the Moral Foreign Language Effect in Russian Speakers. 

We are pleased to hear that the reviewers saw merit in the manuscript, and we have made revisions in accordance with instructions to bring the manuscript into alignment with PLOS One guidelines and standards. Accordingly, we have uploaded a Revised Manuscript with Track Changes, as well as an unmarked version of the original manuscript. We also respond to each point below. 

COMMENT 

Reviewer #2: This paper shows a valuable contribution to the existing literature and knowledge in terms of introduction, style of writing, and methodology. Yet, there are minor typos that should be corrected.

The manuscript should be checked for minor language issues and the researchers need to mention to some tables in the explanations so that they are clearer and easier to be followed. Furthermore, The discussion of the study needs to be compared with more relevant new studies in relation to that period such as:

Liqaa Habeb Al-Obaydi (2021): Humanistic learning elements in a blended

learning environment: a study in an EFL teaching context, Interactive Learning Environments, DOI:

10.1080/10494820.2021.1919717

Finally,it still needs thorough proofreading and editing to polish its content, language, and APA style.

The manuscript looks much better. Thank you.

RESPONSE 

We thank the reviewer for these encouraging words. We went through the manuscript to correct any remaining typos in the text and references and eliminated some phrases with different coloured text—see enclosed version with track changes. 

We now mention tables in the discussion, line 478, line 490, and line 541, in addition to the results. 

We also now cite the mentioned paper as follows: Finally, it may be that the learning conditions under which people encounter their FL influence responses, such as humanistic learning environments [84].

---

## [Decision Letter · Decision Letter 2]

13 Jun 2023

Are People Really Less Moral in Their Foreign Language? Proficiency and Comprehension Matter for the Moral Foreign Language Effect in Russian Speakers

PONE-D-23-05151R2

Dear Dr. Conway,

We’re pleased to inform you that your manuscript has been judged scientifically suitable for publication and will be formally accepted for publication once it meets all outstanding technical requirements.

Kind regards,

Ehsan Namaziandost

Academic Editor

PLOS ONE

Additional Editor Comments (optional):

Reviewers' comments:

Reviewer's Responses to Questions

**Comments to the Author**

1. If the authors have adequately addressed your comments raised in a previous round of review and you feel that this manuscript is now acceptable for publication, you may indicate that here to bypass the “Comments to the Author” section, enter your conflict of interest statement in the “Confidential to Editor” section, and submit your "Accept" recommendation.

Reviewer #2: All comments have been addressed

2. Is the manuscript technically sound, and do the data support the conclusions?

Reviewer #2: Yes

3. Has the statistical analysis been performed appropriately and rigorously? 

Reviewer #2: Yes

4. Have the authors made all data underlying the findings in their manuscript fully available?

Reviewer #2: Yes

5. Is the manuscript presented in an intelligible fashion and written in standard English?

Reviewer #2: Yes

6. Review Comments to the Author

Reviewer #2: Thanks for addressing all the corrections. The authors have adequately addressed the comments raised in a previous round of review and I feel that this manuscript is now acceptable for publication.

7. PLOS authors have the option to publish the peer review history of their article (what does this mean?). If published, this will include your full peer review and any attached files.

Reviewer #2: No

---

## [Editor Report · Acceptance letter]

3 Jul 2023

PONE-D-23-05151R2 

Are people really less moral in their foreign language? Proficiency and comprehension matter for the moral foreign language effect in Russian speakers 

Dear Dr. Conway:

I'm pleased to inform you that your manuscript has been deemed suitable for publication in PLOS ONE. Congratulations! Your manuscript is now with our production department. 

Kind regards, 

on behalf of

Dr. Ehsan Namaziandost 

Academic Editor

PLOS ONE